# Joint Transcriptome and Metabolome Analysis Prevails the Biological Mechanisms Underlying the Pro-Survival Fight in In Vitro Heat-Stressed Granulosa Cells

**DOI:** 10.3390/biology11060839

**Published:** 2022-05-30

**Authors:** Abdul Sammad, Hanpeng Luo, Lirong Hu, Shanjiang Zhao, Jianfei Gong, Saqib Umer, Adnan Khan, Huabin Zhu, Yachun Wang

**Affiliations:** 1National Engineering Laboratory for Animal Breeding, Key Laboratory of Animal Genetics, Breeding and Reproduction, MARA, College of Animal Sciences and Technology, China Agricultural University, Beijing 100193, China; drabdulsammad@outlook.com (A.S.); luohanpeng@cau.edu.cn (H.L.); b20193040324@cau.edu.cn (L.H.); dr.adnan93@cau.edu.cn (A.K.); 2Embryo Biotechnology and Reproduction Laboratory, Institute of Animal Sciences, Chinese Academy of Agricultural Sciences, Beijing 100193, China; zhaoshanjiang@caas.cn (S.Z.); gjf18553676192@163.com (J.G.); 3Department of Theriogenology, Faculty of Veterinary Sciences, University of Agriculture, Faisalabad 38040, Pakistan; saqibumar33@hotmail.com

**Keywords:** granulosa cells, heat stress, integrated analysis, genes, metabolites, protein metabolism, AMPK pathway, cancer pathways

## Abstract

**Simple Summary:**

This study integrates the differentially expressed genes and important metabolites found in the transcriptome and metabolome analyses of in vitro acute heat-stressed bovine granulosa cells. Proteins and vitamin B catabolism were the most; tryptophan, proline, pyridoxine, and riboflavin (downregulated metabolites in heat stress) were enriched in many joint pathways. Genes *AOX1*, *PYGM*, *NOS2*, and *SLC16A3* (all upregulated metabolites in heat stress) were found to have central roles in metabolic changes in response to heat stress. The negative effects of acute heat stress can be attributed to oxidative stress-driven changes at the inflammation metabolism nexus.

**Abstract:**

Previous studies reported the physical, transcriptome, and metabolome changes in in vitro acute heat-stressed (38 °C versus 43 °C for 2 h) bovine granulosa cells. Granulosa cells exhibited transient proliferation senescence, oxidative stress, an increased rate of apoptosis, and a decline in steroidogenic activity. In this study, we performed a joint integration and network analysis of metabolomic and transcriptomic data to further narrow down and elucidate the role of differentially expressed genes, important metabolites, and relevant cellular and metabolic pathways in acute heat-stressed granulosa cells. Among the significant (raw *p*-value < 0.05) metabolic pathways where metabolites and genes converged, this study found vitamin B6 metabolism, glycine, serine and threonine metabolism, phenylalanine metabolism, arginine biosynthesis, tryptophan metabolism, arginine and proline metabolism, histidine metabolism, and glyoxylate and dicarboxylate metabolism. Important significant convergent biological pathways included ABC transporters and protein digestion and absorption, while functional signaling pathways included cAMP, mTOR, and AMPK signaling pathways together with the ovarian steroidogenesis pathway. Among the cancer pathways, the most important pathway was the central carbon metabolism in cancer. Through multiple analysis queries, progesterone, serotonin, citric acid, pyridoxal, L-lysine, succinic acid, L-glutamine, L-leucine, L-threonine, L-tyrosine, vitamin B6, choline, and *CYP1B1*, *MAOB*, *VEGFA*, *WNT11*, *AOX1*, *ADCY2*, *ICAM1*, *PYGM*, *SLC2A4*, *SLC16A3*, *HSD11B2,* and *NOS2* appeared to be important enriched metabolites and genes, respectively. These genes, metabolites, and metabolic, cellular, and cell signaling pathways comprehensively elucidate the mechanisms underlying the intricate fight between death and survival in acute heat-stressed bovine granulosa cells and essentially help further our understanding (and will help the future quest) of research in this direction.

## 1. Introduction

Ovarian granulosa cells present in the follicles nurture oocytes and have specialized roles in steroidogenesis in the ovaries [1,2]. Heat stress has been shown to alter granulosa cell survival, limit cell proliferation transition, decrease hormone synthesis and secretion, cause oxidative stress, and promote apoptotic manifestation [3,4,5]. Impairment of granulosa cells (through heat or occupational hazard stress) leads to disruption of ovarian activity and oocyte development competence [6,7,8]. Higher ambient temperatures are shown to be associated with decreases in ovarian reserves and reproductive aging [9]. Heat stress is also shown to advance the process of luteinization–differentiation in granulosa cells, a condition associated with a decline in fertility [10]. Heat stress causes several metabolic changes in the body [11], where high ketone bodies, high non-esterified fatty acids, and inflammatory changes disrupt the biochemical profiles in ovarian follicles [12,13,14,15,16]. Heat stress-caused changes in the biochemical profiles of ovarian follicles have been associated with dysfunctions in granulosa cells and poor development of oocytes [16,17,18]. Earlier studies about heat stress in granulosa cells reported proliferation senescence and variable potential of granulosa cells to resume proliferation, following acute heat stress exposure [19,20]. In our previous heat stress study on granulosa cells [4], a despaired response was observed, where 40 °C appeared somewhat more lethal than 41 °C. Simulating an in vitro acute heat-stressed (43 °C for 2 h) bovine granulosa cell model could help one grasp the real-time biological insights of cellular stress responses at the molecular level (although it is not representative of actual physiological conditions). Therefore, we performed another comprehensive study of acute heat stress (43 °C) on granulosa cells where transient cellular senescence and later resumption of proliferation were observed in heat-stressed granulosa cells. Our previous transcriptome level studies involving an extensive range of temperatures gave useful biological insights into the cellular mechanisms, metabolic level changes, and apoptotic and antioxidant pathways [4,21]. In our LC–MS-based untargeted metabolomics study, important metabolites, metabolic pathways, and their interplay in cellular mechanisms were reported in acute heat-stressed bovine granulosa cells [22]. Our transcriptome study reported evidence of an intricate fight between pro-survival and pro-death pathways, while the metabolome study suggested differential metabolites directed towards bioenergetic support mechanisms in acute heat-stressed bovine granulosa cells [21,22]. Integrated transcriptome and metabolome analyses involving correlation and network analyses are efficient strategies to explore the mechanisms underlying a biological system. Therefore, this study evaluates cellular physical responses and integrates the transcriptome and metabolome data from acute heat-stressed bovine granulosa cells [21,22] to narrow down and explore new biological insights regarding important genes, metabolites, and adaptive pathways involved in the aftermath of acute heat stress to bovine granulosa cells.

## 2. Materials and Methods

### 2.1. Granulosa Cell Culture, Heat Treatment, and Cell Experiments

The cell collection, culture, and treatment methods followed our previous study [21,22,23]. In brief, follicular fluid was collected from small follicles (3–8 mm) of cyclic bovine ovaries followed by filtration and centrifugal washings. Obtained cells were seeded on coverslips in a 12-well plate DMEM/F12 medium containing 10% FBS (both from Thermo Fisher Scientific, Waltham, MA, USA). After 24 h of culturing, cells were fixed, permeabilized, washed with PBS, loaded with 5% goat serum for 1 h, and incubated with the FSHR antibody (1:1500, Sigma Aldrich, St. Louis, MI, USA) at 4 °C overnight. In the next step, cells were subjected to FITC-conjugated secondary antibody for 2 h, followed by washing and PI staining (Nanjing Jiancheng Bio Inst., Nanjing, China). Finally, slides were observed under a fluorescence microscope (Olympus, Tokyo, Japan); the results are shown in Appendix A. The obtained FSHR-positive granulosa cells were initially cultured for 48 h and subsequently subjected to 2 h of acute heat stress (43 °C) in a humidified 5% CO_2_ incubator, while the control group of cells remained at 38 °C.

Intracellular reactive oxygen species (ROS), apoptosis, and hormone measurement methods followed our previous studies [21,22,23]. Cells were cultured at the rate of 2 × 10^4^ for a colorimetric CCK-8 kit (Dojindo Laboratories, Kumamoto, Japan) based on cell viability in each well (of 96-well plates). Post-treatment absorbance of cells was measured at a 450 nm wavelength by a plate reader for cell viability, followed by similar repeated measurements at various time intervals (cell proliferation). Similarly, 1 × 10^4^ cells in each well (of 96-well plates) were cultured for fluorocolorimetric ROS assay (DCFDA kit, Abcam, Cambridge, MA, USA). Post-treatment fluorescence absorbance of cells was measured at excitation/emission = 485/535 nm by a plate reader for the ROS assay. Using the Annexin V-FITC kit (Nanjing Jiancheng Bio Inst., Nanjing, China), Annexin V-FITC /PI double staining was performed to evaluate the granulosa cell apoptosis according to the manufacturer’s instructions before being analyzed by flow cytometry. The data were analyzed by Flowjo software (version Win64–10.4.0). Similarly, the determinations of P4 and E2 concentrations in treated cell culture media were measured according to the manufacturer’s protocols of the respective ELISA kits (Cusabio Technology LLC, Wuhan, China). The data from at least six replicates each for cell proliferation, ROS, apoptosis measurements, and ELISA were analyzed using Graphpad Prism version 9.0.0 for Windows (GraphPad Software, Inc., San Diego, CA, USA). Analysis of variance was carried out and means were compared through Tukey’s honestly significant difference (HSD) test at a 5% level of significance (α = 0.05). All the data presented in the figures are expressed as means ± standard errors (S.E).

### 2.2. Transcriptomics Data and Differentially Expressed Genes

Comprehensive methods of RNA sequencing were explained in our previous articles [21,22]. In brief, the RNA was isolated from the control (38 °C) and heat-stressed (43 °C for 2 h) bovine granulosa cells according to the TRIzol Reagent method. RNA quality, integrity, and concentration were assessed, followed by the cDNA library with the NEBNext Ultra RNA Library Prep Kit for Illumina (cat no. E7530S, New England Biolabs (UK) Ltd., Hitchin, Herts, UK) and finally submitted for sequencing by the NovaSeq 6000 System (Illumina, Inc., San Diego, CA, USA), which generated 150 base paired-end reads. The quality check (FastQC software (v0.11.9) and global trimming (Fastp, v0.20.0) [24] reads were mapped to the bovine genome of version ARS-UCD1.2. Gene expression counts were investigated through RNA-SeQC software (v2.3.6) [24]. Principal component analysis (PCA) and clustering structure were performed using the psych and hcluster R packages. For differentially expressed gene (DEG) screening, quantile-adjusted conditional maximum likelihood (qCML) was performed using edgeR in the R package [25] with criteria logFC ≥ 1.5 and 0.05 for the alpha of false discovery rate (FDR). Finally, 330 significant DEGs were determined in the control versus heat-stressed bovine granulosa cells group comparisons.

### 2.3. Metabolomics Data and Differentially Expressed Metabolites

Comprehensive methods involved in the metabolome assay were detailed in our previous study [22]. Briefly, cell culture fluid was collected from the control (38 °C) and heat stress (43 °C for 2 h) groups. After the initial methanol extraction, 3 replicate samples from each group were dispatched for liquid chromatography–tandem mass spectrometry (LC–MS/MS) analysis. Further, metabolite extraction was done through ice-cold acetonitrile; samples were air-dried and resuspended in pure water. All extracted samples and quality controls were run through LC–MS/MS using HSS T3 100 ×2.1 mm 1.8 μm column (Waters) on Ultimate 3000 (Thermo Fisher Scientific, Waltham, MA, USA) followed by an analysis employing the Q Exactive system (Thermo Fisher Scientific, Waltham, MA, USA). Qualitative results of samples and peak intensities in both ion modes of LC–MS analysis were obtained and the MSBank and KEGG databases were queried for metabolites identification. The MetaboAnalyst 5.0 package [26] was employed to carry out the PCA for investigating the clustering trends and outliers. Moreover, metabolites with variable importance in projection (VIP) values greater than 1 (with their fold change values), calculated through the partial least-squares discriminant analysis (PLS-DA), were considered the differential metabolites between the control and heat-stressed groups. Finally, a list of 56 important differential metabolites was determined in the control versus heat-stressed bovine granulosa cell group comparisons.

### 2.4. Integrated Pathway Analysis of Genes and Metabolites

Henceforth, in this study, we conducted a joint integrative analysis of our previous metabolome [22] and transcriptome data from in vitro acute heat-stressed (43 °C for 2 h) bovine granulosa cells. Both of those earlier omics studies involved the same samples and replicates. Out of 330 significantly differentially expressed genes (FC ≥ 1.5 and 0.05 for the alpha of FDR), a total of 256 genes (along with their log2(FC) values) were finally mapped to the different KEGG databases used by the latest version of software MetaboAnalyst 5.0 [26]. Similarly, a list of 56 important metabolites (PLS-DA, VIP score ≥ 1) along with the log2(FC) values were finally used in a joint integrative analysis of metabolomics and transcriptomic data in MetaboAnalyst 5.0 software [26]. In the first step of the analysis, the “Joint-Pathway Analysis” module of MetaboAnalyst 5.0, based on the KEGG global metabolic network (ko01100), was used. Mapping of both gene and metabolite features was assessed and manually corrected in case matching options were available. The first step of analysis involved the following sub-modules, “All pathways (gene only)”, “Metabolic pathways (metabolites only)”, “All pathways (integrated)”, and “Metabolic pathways integrated”. The default settings were maintained as the enrichment analyses based on the “Hypergeometric test”; the topology measure as “Degree centrality”; and the integration method of “Combined queries”. Dot plots, tables of pathway results, and matched features were saved and downloaded for each analysis. Additionally, the module “All pathways (integrated)” was used for visualization of the network among pathways and enriched features.

### 2.5. Interaction Network Analysis among Genes and Metabolites

For this purpose, the “Joint-Pathway Analysis” module of MetaboAnalyst 5.0 was used. Mapping of both gene and metabolite features was assessed and manually corrected in case matching options were available. In the first step, the “Gene-metabolite interaction network” sub-module with default settings was used for the interaction analysis among genes and metabolites. In the second step, the “Metabolite-metabolite interaction network” sub-module with default settings was used for the interaction analysis among metabolites based on both genes and metabolite data input. For the third interactive network analysis, the “Metabolite-gene-disease interaction network” sub-module with default settings was used for the interaction analysis among genes. Moreover, all of the important DEGs found in the integration and network analyses of metabolome and transcriptome data were subjected to a PPI physical network analysis at a medium confidence score of 0.4 using STRING version 11.5 (https://cn.string-db.org/ accessed on 22 April 2022).

## 3. Results

### 3.1. Effect of Heat Stress on Granulosa Cell Parameters

Bovine granulosa were exposed to in vitro heat stress treatment (43 °C) while the control group remained at 38 °C [21,22]. Cells in the control group maintained steady proliferation activity (Figure 1A); no change was observed until 24 h in the treatment group (Figure 1A). A significant (*p* < 0.05) difference in cell viability was observed at 24 and 48 h time points between both groups. A significant increase in the intracellular ROS level was observed in the treatment group compared to the control group (Figure 1B). Similarly, heat-stressed GCs had a significantly higher (*p* < 0.05) apoptotic rate (Figure 1C), as shown in the representative flow cytometry-based apoptotic measurements in the control (Figure 1D) and heat stress (Figure 1E) groups. Similarly, E2 and P4 levels were significantly (*p* < 0.05) decreased in the culture media of the heat-stressed GCs compared to the control GCs (Figure 1F,G, respectively).

### 3.2. Transcriptome and Metabolome from Heat-Stressed Granulosa Cells

A total of 256 significant DEGs and 51 differentially expressed important metabolites along with their logFC values were found in the RNA sequencing analysis and LC–MS-based untargeted metabolome analysis, respectively, which were used in this study for integrated analysis of transcriptome and metabolome underlying the biological mechanisms of acute heat-stressed (43 °C for 2 h) bovine granulosa cells. These data came from our previous transcriptome and metabolome investigation performed on the same sample replicates of the acute heat-stressed granulosa cells [21,22]. The PCA score plots of the transcriptome data and the PLS-DA score plots for metabolites are given in Appendix A. The 97 significantly differentially expressed genes and metabolites as well as their logFC values used in this study are provided in Appendix A.

### 3.3. Integrated Pathways Analysis of Genes and Metabolites

#### 3.3.1. Joint Pathway Enrichment of Genes Used in This Study

Differentially expressed genes of the transcriptomic data used in the integrated analysis of this study were subjected to the Kyoto encyclopedia of genes and genomics (KEGG) pathway-based enrichment analysis; results of the pathway analysis are illustrated in Figure 2.

Statistical results of the transcriptomic-enriched pathway analyses and a list of enriched genes in each pathway are given in Appendix A. Out of the functional pathways, TGF-beta and VEGF signaling pathways were important; numerous cellular processing pathways include relaxin signaling, cytochrome P450 pathways, apelin signaling, circadian entrainment, chemical carcinogenesis, dilated cardiomyopathy, and axon guidance pathways enriched among bovine granulosa cells in response to acute heat stress. A large number of metabolic pathways were enriched in this integrative analysis, out of which amino acid metabolic pathways were numerous, including tyrosine metabolism, arginine biosynthesis, glycine, serine, threonine metabolism, and tryptophan metabolism. In carbohydrate metabolism pathways, the starch and sucrose metabolism pathways were enriched. The last important component of the integrative analysis included lipid metabolism, where linoleic acid metabolism, ether lipid metabolism, and arachidonic acid metabolism pathways were enriched. Similarly, all differential metabolites of metabolome data were also subjected to the KEGG-based pathway enrichment analysis, as shown in the illustration in Figure 3; the statistical details of the analysis and the list of enriched metabolites are given in Appendix A. In only the metabolite pathway analysis, aside from aminoacyl-tRNA biosynthesis, major enriched pathways were composed of amino acid metabolism. Other important pathways were composed of vitamin B6 metabolism, riboflavin metabolism, and citrate (TCA) cycle.

#### 3.3.2. Combined Pathway Enrichment of Genes and Metabolites

Significant DEGs and important metabolites found in heat-stressed granulosa cells were subjected to the integrated KEGG-based pathway enrichment analysis, as illustrated in Figure 4. As shown in Figure 4, ABC transporters, central carbon metabolism, aminoacyl-tRNA synthesis, protein digestion and absorption, and glycine, serine, and threonine metabolism pathways were significantly enriched in response to heat stress treatment in bovine granulosa cells. Similarly, the pathways with the most biological impacts in the molecular processes employed by granulosa cells to acute heat stress were composed of one carbon pool by folate, vitamin B6 metabolism, and arginine and tyrosine metabolism pathways.

Statistical summary results of the top enriched pathway of the integrated analysis are detailed in Table 1; the detailed results of all enriched pathways along with the list of enriched genes and metabolites in each of these pathways are listed in Appendix A.

#### 3.3.3. Joint Metabolic Pathway Analysis

Several important metabolites found in the metabolomes of heat-stressed granulosa cells are few (in number). A joint pathway analysis of data involving only metabolic pathways was carried out (Figure 5). The statistical details and enriched genes or metabolites found in each of the enriched metabolic pathways are given in Appendix A. Results of this analysis provided additional insight into the granulosa cells’ metabolic pathways employed in response to acute heat stress, where additional metabolic pathways observed included arginine and proline metabolism and drug metabolism by cytochrome P450; alanine, aspartate, and glutamate metabolism were found.

### 3.4. Important Pathways, Metabolites, and Genes in the Joint Pathway Analysis

A joint analysis of metabolome and transcriptome data (under Section 3.3.2) was queried for significant and interesting functional pathways related to cellular functions and adaptation of granulosa cells to heat stress (Table 2).

Furthermore, a network analysis of all pathways was carried out using transcriptome and metabolome data input. Figure 6A presents the comprehensive network of the integrated pathway network, where all mapped queries of metabolites involved in pathways with more than one hit are labeled. Similarly, Figure 6B shows the same network, with labeled metabolite nodes involved in multiple pathways with more than one hit. Complete statistical details of this integrated network analysis of all pathways and enriched genes and metabolites are presented in Appendix A.

Additionally, a joint metabolic analysis of transcriptome data (under Section 3.3.3) was queried for significant and interesting metabolic pathways related to cellular functions and adaptation of granulosa cells to heat stress (Table 3).

### 3.5. Interaction Network Analysis among Transcriptome and Metabolome Data

#### 3.5.1. Interaction Network Analysis among Genes and Metabolites

Significantly differentially expressed genes and metabolites were used to carry out the KEGG-based interactive network analysis among them, as shown in the interaction network illustration in Figure 7, where important nodes are highlighted. Statistical details of joint interactions of top interactions and those involving two or more nodes along their regulation statuses in the respective studies are presented in Table 4; the complete statistical details of this network are given in Appendix A. 

#### 3.5.2. Interaction Network among Metabolites

A joint interaction network among metabolites based on the KEGG database is illustrated in Figure 8. Important central metabolites labeled in the network illustration are statistically detailed along with their regulation statuses in heat-stressed granulosa cells, in Table 5. The statistical details of the whole interaction network are given in Appendix A.

#### 3.5.3. Interactive Network Analysis among Genes, Metabolites, and Diseases

Significantly differentially expressed genes and metabolites were used to carry out the KEGG-based interactive disease network analysis, as shown in the interaction network illustration in Figure 9, where important disease-, gene-, and metabolite nodes are highlighted. Statistical details of the joint interactions of top disease interactions, metabolites, and genes enriched along their regulation statuses are presented in Table 6; the complete statistical details of this network are given in Appendix A. 

To obtain functional insights into the DEGs found in multiple integrations and the network analysis of transcriptomes and metabolomes, all DEGs were combined and a physical protein–protein interaction (PPI) network analysis (STRING) with a medium confidence score (0.4) was carried out, as presented in Figure 10. Based on experimentally verified–curated database interactions, and text mining, four subnetworks were observed. *GNB3*, *KDR*, *CYP1B1*, *ADCY2*, *RGS9*, *INSRR*, *GSTA5*, *VEGFA*, *ADH6*, *IGF2,* and *MDM4* were the central network nodes.

## 4. Discussion

Heat stress caused by biochemical alterations in the maternal reproductive system has been attributed to low conception rates and higher embryonic losses in cattle [12,27,28]. Granulosa cells present in the ovarian follicles are important for oocyte development due to steroidogenesis and cross-talk with oocytes [1,2]. In addition to our studies [4,21], earlier studies extensively reported the adverse effects (at varying degrees) of heat stress on granulosa cells [3,19,29]. Heat stress promotes oxidative stress and inflammatory cytokine signaling; there is evidence of an intricate survival fight in the presence of pro-apoptotic and anti-apoptotic genes and energetic support pathways [4,21]. This study furthers our knowledge of these complex mechanisms of survival and death.

A pronounced decline of cell proliferation potential was observed in heat-stressed granulosa cells. Earlier studies also reported similar results, where acute heat-stressed (45 ℃) granulosa cells regained proliferation by 48 h after the completion of heat treatment [19,20,30]. However, the 48 h trend of cell proliferation activity recovery followed the results reported by other studies in granulosa cells [3,31]. Interestingly, earlier we reported that immediate culture media change after the completion of heat stress induced early cell proliferation when compared to the heat stress group with culture medium change at 48 h post-treatment [21]. This phenomenon needs further investigation; a preliminary hypothesis may be drawn that heat stress is an energy-intensive process, as supported by our previous review of the literature studies [8,32,33]. Acute heat stress accumulated intracellular ROS and caused an increase in apoptosis of granulosa cells; this phenomenon is commonly observed in heat-stressed cells [4,29]. Certain levels of ROS keep producing due to fundamental biological processes of cells [34]. Nevertheless, higher intracellular ROS in the presence of stress and mitochondrial damage [3] impairs cell antioxidant response elements (ARE) and causes oxidative stress [4,35]. This oxidative stress may ultimately induce granulosa cell apoptosis and compromise the optimal ovarian functions [36,37]. The stress response pathway, the Nrf2 pathway, is particularly involved in diverse up-regulatory functions related to cell metabolism in conjunction with nuclear factor kappa-light-chain-enhancer of activated B cells (NF-κB) [38,39], PI3K/AKT/mTOR [40,41,42], and AMPK signaling pathways [43]. This integrative analysis of metabolome and transcriptome data indicates high amino acid metabolism pathways and metabolites, of which, tyrosine, tryptophan, threonine, phenylalanine, and arginine metabolism pathways are enriched, indicating the implications of these pathways. Nrf2 and NF-κB pathways are supportive and sometimes antagonistic to bringing up cellular homeostasis in the event of stress; NF-κB promotes inflammation and inflammation and subsequently promotes transcription activities related to energetic catabolic support [44]. The complex role of the transforming growth factor-β (TGF-β) pathway is also important in this context; it is involved in signaling mechanisms related to the upregulation of NF-κB, inducing apoptosis through cell cycle arrest [45,46]. Our integrative analysis supports this discussion where we found TGF-β and VEGF signaling pathways implicated in heat-stressed granulosa cells. Regarding the involvement of the AMPK signaling pathway, our integrative analysis provides enough evidence of its most versatile involvement, as depicted by *PYGM*, *IGF2*, *MAOB*, *INSL3*, *AOC2, AOX1*, *ATP2A1*, *SLC2A4*, *SLC27A5*, *ADH6*, *NOS2*, etc. These genes are particularly involved in catabolic support to heat-stressed granulosa cells, as shown by starch and sucrose metabolism and various amino acid metabolism pathways.

ATP-binding cassette (ABC) transporters constitute a superfamily of conserved membrane proteins responsible for the transport of nutrients across cells [47]. This pathway is interesting due to its role in cholesterol and lipid transport [48]; earlier we reported on the low steroidogenic activity of heat-stressed granulosa cells despite the evidence of upregulation of cholesterol synthesis in heat-stressed granulosa cells [21,22]. The protein digestion and absorption pathway was the second most significant pathway enriched in the integrated analysis, where leucine, glutamine, arginine, and tyrosine were upregulated in heat-stressed granulosa cells. Yielding of amino acids—and possibly their catabolism in heat-stressed granulosa cells, in conjunction with high inflammatory response, upregulation of the Nrf2 pathway, and *SLC7A11* [49]—may be a protective strategy of heat-stressed granulosa cells [21,22]. Another important pathway where metabolites and genes converge is the cAMP signaling pathway. Mitochondrial stress, oxidative stress, and the cAMP signaling pathway nexus [50] appear to be correlated in heat-stressed granulosa cells [3,4]. Furthermore, the cAMP signaling pathway activates complex signaling involving the MAPK pathway, which can ultimately promote DNA damage and cell senescence [51]. The most important integrated enrichment pathway may probably be ovarian steroidogenesis. This pathway enrichment supports our earlier findings of low progesterone synthesis by heat-stressed granulosa cells [4,21]. In our integrated analysis, the enrichment of the mTOR pathway is reported with four hits, including upregulation of leucine and arginine and downregulation of WNT11 and DEPTOR. The mTOR pathway is shown to downregulate in cell cycle arrest and its duopoly with the IGF2 pathway should be investigated further at metabolic and cell senescence levels in heat-stressed granulosa cells [52,53]. The most interesting evidence is of the AMPK signaling pathway, characterized by upregulation of GLUT4. Regarding the AMPK signaling pathway in our transcriptomic study [21], it is characterized by downregulation of the ATP gene family and *PPARGC1A* and upregulation of *SIRIT1, SLC2A4, HMGCS1*, and *SERBP1* [43,54]. Glutamine and citric acid, both upregulated metabolites, were found as the most abundantly enriched metabolite nodes in the integrated interaction network analysis of both omics data. Both metabolites were enriched in many pathways and appear as the most important markers of the heat-stressed phenotype of granulosa cells.

A joint network analysis of metabolites and genes revealed *CYP1B1*, *PYGM*, *ICAM1*, *HSD11B2*, and *OXT*. Downregulation of *HSD11B2* is justifiable as there is evidence of the low steroidogenic capacity of heat-stressed granulosa cells [4,21]; however, we had additional strong evidence of the transcriptional suppression of the steroidogenic potential of heat-stressed granulosa cells in the form of genes observed in the integrated analysis, such as *ADCY2*, *CYP1B1*, and *PLA2G4B*. This decline in the steroidogenic activity of GCs can be attributed to the transcriptional downregulation of *STAR*, *CYP11A1* [4], and mitochondrial damage due to ROS [3]. This study could not obtain deep mechanisms regarding steroidogenesis modulation in heat-stressed granulosa cells; however, our results present a further line of investigation. The established phenomenon of blunt non-esterified fatty acid response in heat-stressed cows was characterized by low-fat mobilization [11,55,56]; upregulated genes involved in fatty acids and cholesterol metabolism indicates that fatty acid metabolism is an adaptation of heat-stressed granulosa cells. It can be concluded that fat mobilization is altered at the organism level in heat stress, but the cellular metabolism of fats remains enhanced. Therefore, supplemental fats can help to avert the heat-stressed typical negative energy balance and associated adverse effects [57]. Furthermore, upregulated *ICAM1* and *PYGM* are probably interesting findings. Amino acid catabolism and altered immune response in the shape of upregulated cytokine signaling may explain its role in heat-stressed granulosa cell adaptation and cancer cells. The highly upregulated *PYGM* gene, an enzyme involved in glycogenolysis, is involved in diverse insulin and glucagon signaling, necroptosis, inflammatory response, cellular energy support, and cancer progression [58]. Similarly, upregulated *ICAM1* in the one-carbon metabolism pathway through an integrated analysis is an important finding because its upregulation reduces ovarian cancer cell growth [59]; at the same time, it needs further investigation because our transcriptome data suggested high inflammatory responses, in the forms of upregulated TLRs, cytokines, Fas, and Fasl [59].

Four cancer pathway enrichments were observed in this study, characterized by the upregulation of glutamine, citric acid, leucine, arginine, tyrosine, and the downregulation of progesterone. These findings are related to cancer, as heat-stressed cells resemble the cancer phenotype; at the same time, heat treatment is a potential therapy for the treatment of tumors. Heat treatment has been long regarded as an important treatment option for cancer [60]; however, there are cons involving heat treatment, e.g., studies have reported multipolar divisions in cancer cells, ultimately facilitating tumor heterogeneity [61]. An interplay of the energy metabolic dynamism was reported in hyperthermia-resistant ovarian cancer cells [62]. Metabolic changes are shown to play decisive roles in heat treatment-based cell death [63]. The most important gene found in cancer pathways through integrated analysis is *SLC16A3*, a monocarboxylate transporter [64]. This gene is involved in energetic metabolism through fatty acid metabolism and has a connected role in the citrate (TCA) cycle [64], which is complicated in cancer through the p53 and mTOR pathways [64,65]. The upregulated status of *SLC16A3* is particularly shown as a prognostic biomarker of cancer aversion [64,65]; its upregulation in heat-stressed granulosa cells is notable. Little apoptosis in a cell senescence scenario causes malignancy in the cells; apoptosis pathways are complex and problems at any step can hinder cancer treatment. Targeting apoptosis will remain the main cancer treatment strategy [66]. In this integrative analysis, we observed *MDM4* and *CSNK1B* being interconnected as well as nodes toward the involvement of the p53 signaling pathway at the conjunction of metabolic pathways through complex transcription mechanisms. Our transcriptomics study (reporting upregulation of p21 and p16 and downregulation of p53) is interesting, suggesting cell cycle arrest in heat-stressed granulosa cells [21], and may lead to the cellular senescence phenotype [67]. In the absence of p53 and associated *MDM4* activity, and upregulated p21 and p16, a fight between cell senescence and apoptosis can be assumed [67,68,69]. Senescent cells remain viable and alter metabolic and gene expression profiles [67]. Metabolic alterations related to acetyl-CoA, citrate cycle, and amino acid catabolism are evident in our metabolome data of heat-treated granulosa cells [22]. Recently, dysregulation of amino acids, such as tryptophan, phenylalanine, and tyrosine, were implicated in colorectal cancer [70]. Cell cycle arrest can also induce cell apoptosis, where along with co-expression of p21 and p16, p53 appears as a double-edged sword [71]. Heat stress is shown to alter the morphology of granulosa cells [19] and upregulate DNA damage and genes related to structural remodeling [3,21]. These phenomena can ultimately lead to activation of p53, inducing cell-cycle arrest, apoptosis, senescence, and DNA repair [72,73]. High ROS activity, downregulated *MDM4*, *IGF2*, *PPARGC1A*, *PPARGC1B*, and upregulated *HIF1A* suggest the interconnecting role of p53 and heat-stressed mitochondrial damage [72,74], as shown in an in vitro heat stress treatment study on mammary epithelial and granulosa cells [75,76].

## 5. Conclusions

The etiology of various changes in heat-stressed granulosa can be (at least in part) attributed to varying degrees of cellular stress, impairment in lipid transport and metabolism, the inflammation–metabolism nexus, and disruption in energetic metabolism. Threonine, phenylalanine, and proline appeared to be the only downregulated abundant amino acids in various pathways. Similarly, the involvement of downregulated succinic acid, vitamin B6, and riboflavin in many pathways are important as well as nodes toward the simple sugars and vitamin importance in the fight against heat stress at the cellular level. Since we observed cell proliferation activity recovery after heat stress, upregulated choline and citric acid metabolites may primarily be considered important metabolites involved in heat-stressed granulosa cells. Downregulated *MAOB*, *AOC2,* and upregulated *AOX1*, *ADH6* pairs involved in vitamin B6 and amino acid metabolism may be considered important candidate genes. This study provided additional evidence of the transcriptional suppression of the steroidogenic potential of heat-stressed granulosa cells in the forms of *ADCY2*, *CYP1B1*, and *PLA2G4B*; *COL11A2* (a downregulated gene) may be considered a universal candidate gene involved in amino acid metabolism. Furthermore, upregulated *ICAM1* and *PYGM* are likely interesting findings regarding the inflammation metabolism nexus in granulosa cells exposed to heat stress. Moreover, via the demonstrations of various cell signaling and metabolic pathways, this integrative analysis of metabolome and transcriptome acute heat-stressed granulosa cells provided an interesting conjecture of metabolites and genes through an integrated analysis of transcriptome and metabolome data.

## Figures and Tables

**Figure 1 biology-11-00839-f001:**
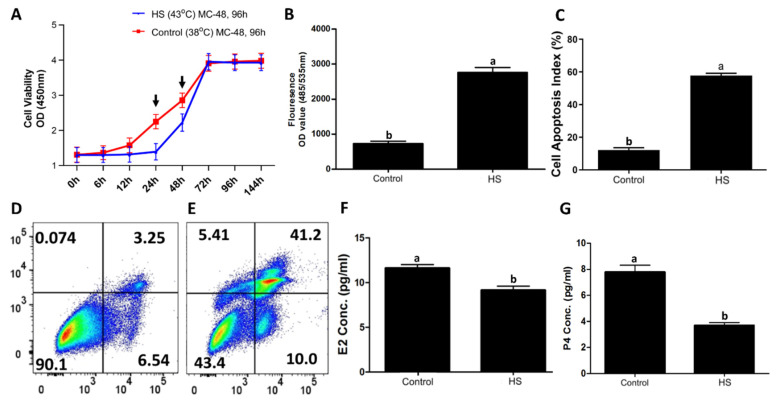
In vitro heat stress (43 °C for 2 h) versus the control (38 °C) treatment decreased cell viability, increased intracellular ROS accumulation, increased the incidence of apoptosis, and decreased the estrogen (E2) and progesterone (P4) syntheses in bovine granulosa cells [21,22]. Heat stress exposure decreased granulosa cell viability (**A**) Cell proliferation curves with mean optical densities (ODs) were plotted against different recovery time points in hours (h) for both groups. Heat stress exposure increased intracellular ROS accumulation in granulosa cells (**B**) fluorocolorimetric OD values of ROS in cells are shown (**B**). Heat stress exposure elevated granulosa cell apoptosis and decreased viability (**C**) Flow cytometry analysis of granulosa cells cultured under heat stress (**E**) and the corresponding control (**D**) are presented. Heat stress exposure decreased E2 (**F**) and P4 (**G**) synthesis by granulosa cells (**A**). All data are presented as means ± S.E. Time points (24 and 48 h) with black arrow signs above them are significantly (*p* < 0.05) different (**A**), while “MC” refers to the culture media change at the given hours (**A**). Means without common letters (a, b) are significantly different (*p* < 0.05).

**Figure 2 biology-11-00839-f002:**
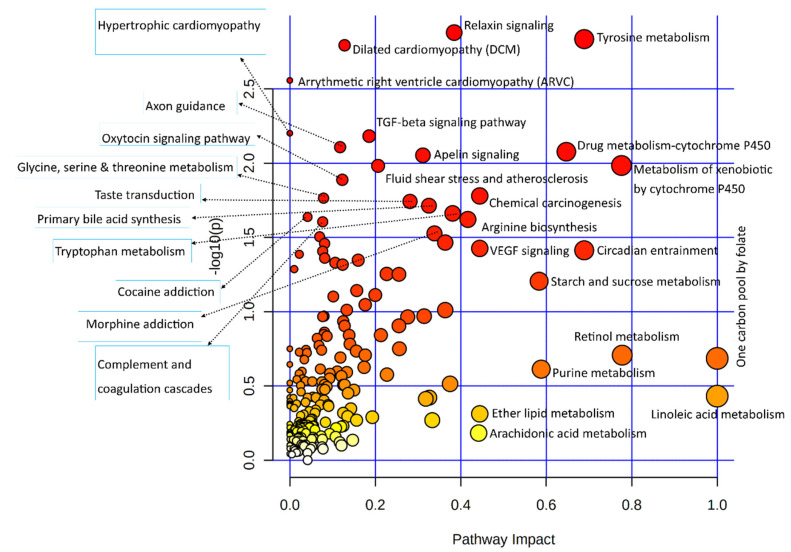
Dot plot illustration of the pathway enrichment of significantly differentially expressed genes used in this study. Various pathways in the upper and left quadrants are labeled. The dot sizes increase with the increasing pathway impact; color intensifies according to the *y*-axis.

**Figure 3 biology-11-00839-f003:**
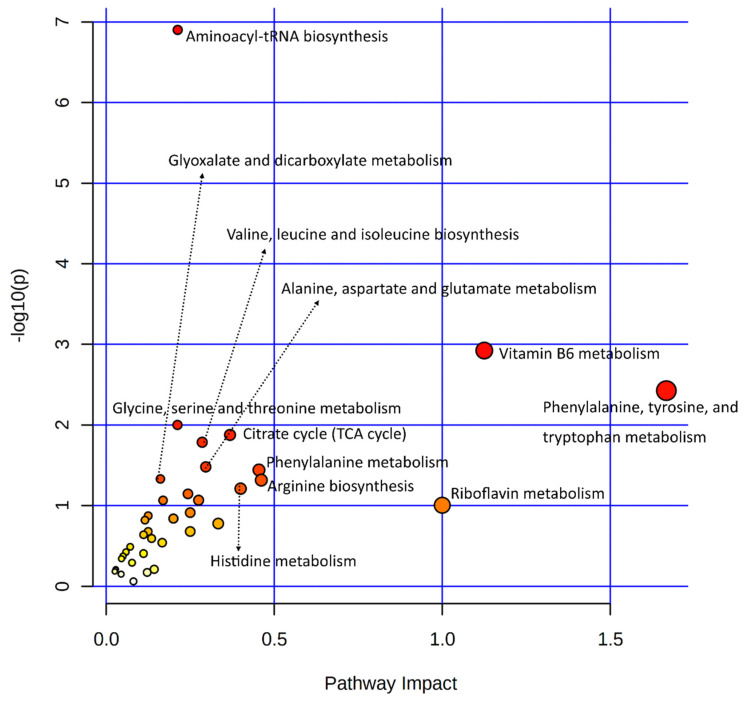
A dot plot illustration of pathway enrichment by significantly differentially expressed metabolites was used in this study. Various pathways within the upper and left quadrants are labeled. The dot sizes increase with the increasing pathway impact; the color intensifies according to the *y*-axis.

**Figure 4 biology-11-00839-f004:**
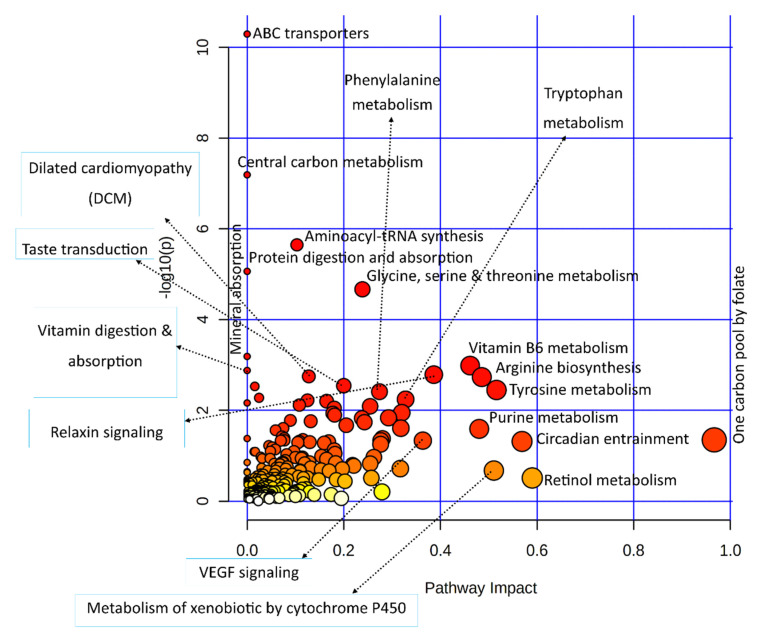
A dot plot illustration of pathway enrichment in the integrated analysis of significantly differentially expressed genes and important metabolites from transcriptome and metabolome data, respectively. Various pathways in the upper and left quadrants are labeled. The dot sizes increase with the increasing pathway impact; the color intensifies according to the *y*-axis.

**Figure 5 biology-11-00839-f005:**
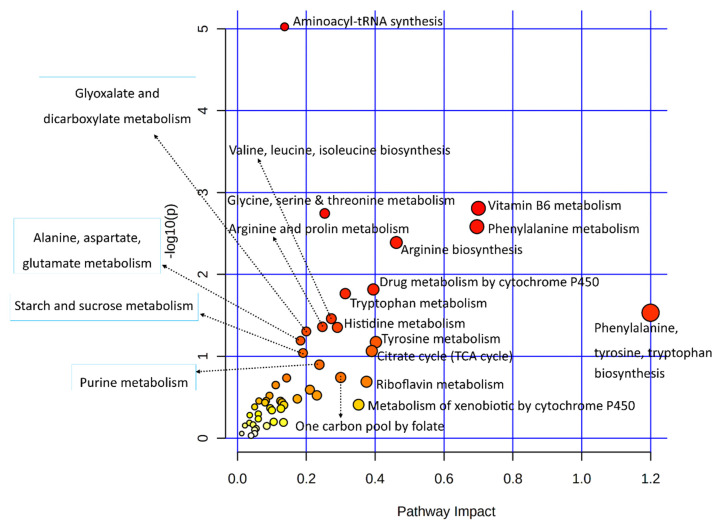
A dot plot illustration of the metabolic pathway enrichment in a joint analysis of significantly differentially expressed genes and important metabolites from transcriptome and metabolome data, respectively. Various metabolic pathways in the upper and left quadrants are labeled. The dot sizes increase with the increasing pathway impact; the color intensifies according to the *y*-axis.

**Figure 6 biology-11-00839-f006:**
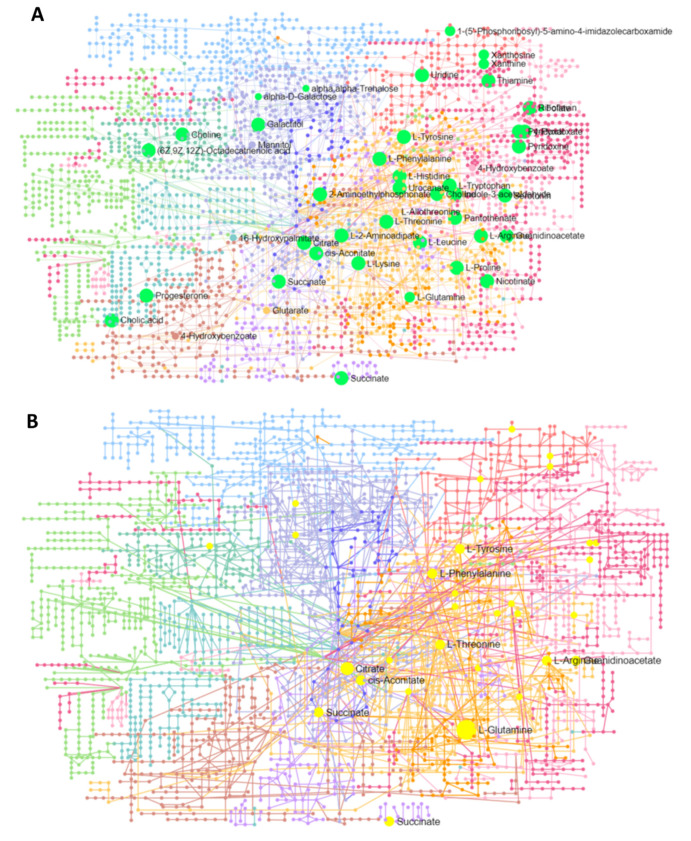
A network analysis of pathways carried out using an integrated analysis of all pathways (using transcriptome and metabolome data input); (**A**) presents the comprehensive network of integrated pathways, where all mapped queries of metabolites (highlighted in light green) are involved in pathways with more than one hit are labeled. Similarly, (**B**) shows the same network, with metabolite nodes involved in multiple pathways (highlighted in yellow) with more than one hit being labeled.

**Figure 7 biology-11-00839-f007:**
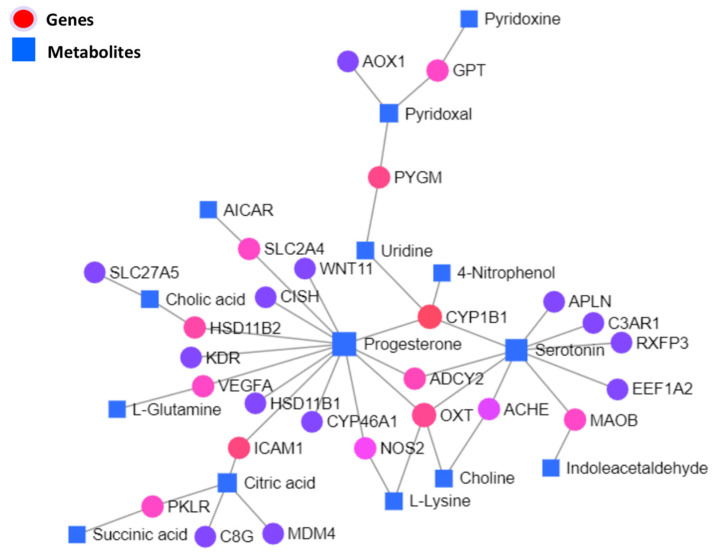
Integrated network illustration of differential genes and metabolites based on the KEGG database. Where circular nodes represent genes, square nodes represent metabolites as labeled. Color variations of gene nodes from red to violet are based on the degree of connections in the network.

**Figure 8 biology-11-00839-f008:**
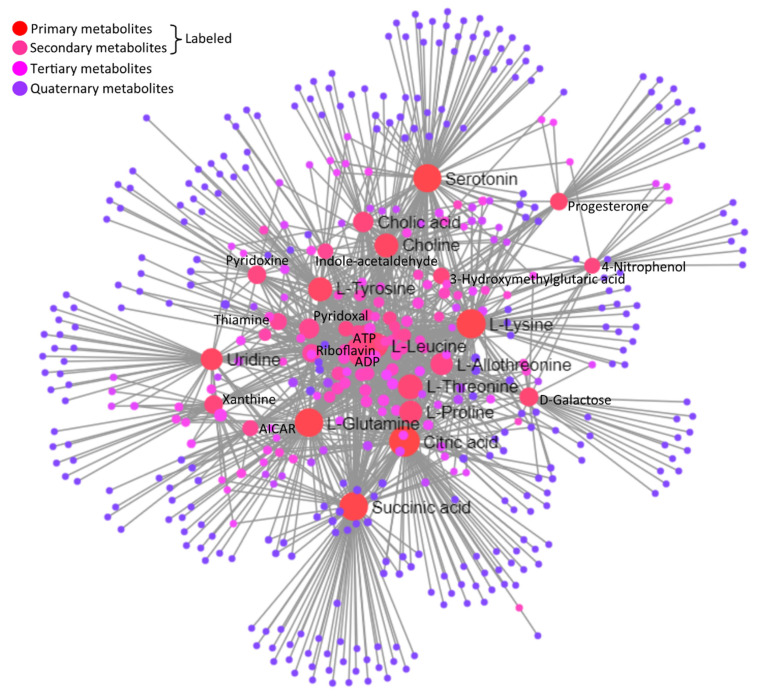
Integrated network illustration of differential metabolites based on the KEGG database. Circular nodes represent different metabolites at different degrees of interaction made through lines connecting and originating them. Central important metabolites in the network are labeled.

**Figure 9 biology-11-00839-f009:**
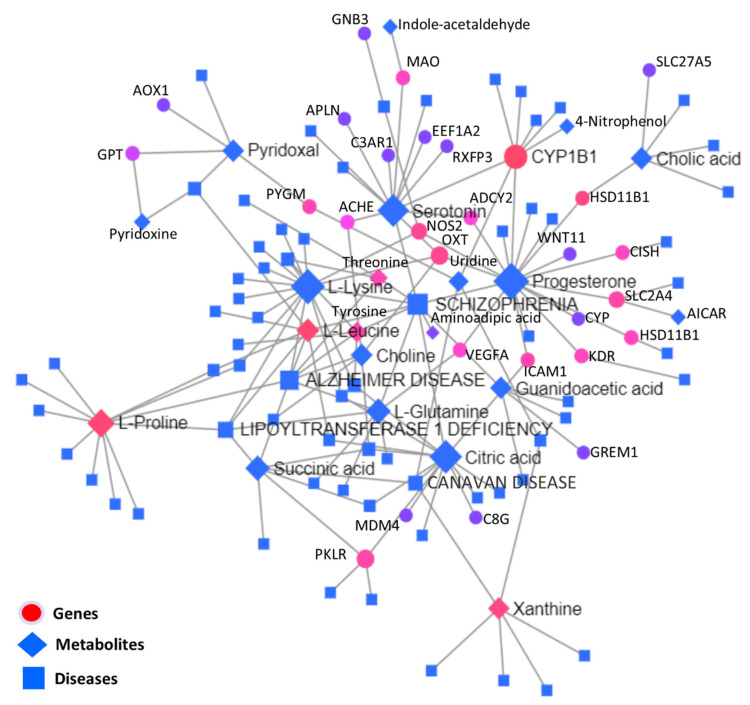
KEGG-based interaction network analysis among metabolites, genes, and diseases, where different shapes of nodes denote different characters, and the lines between different nodes show the connections.

**Figure 10 biology-11-00839-f010:**
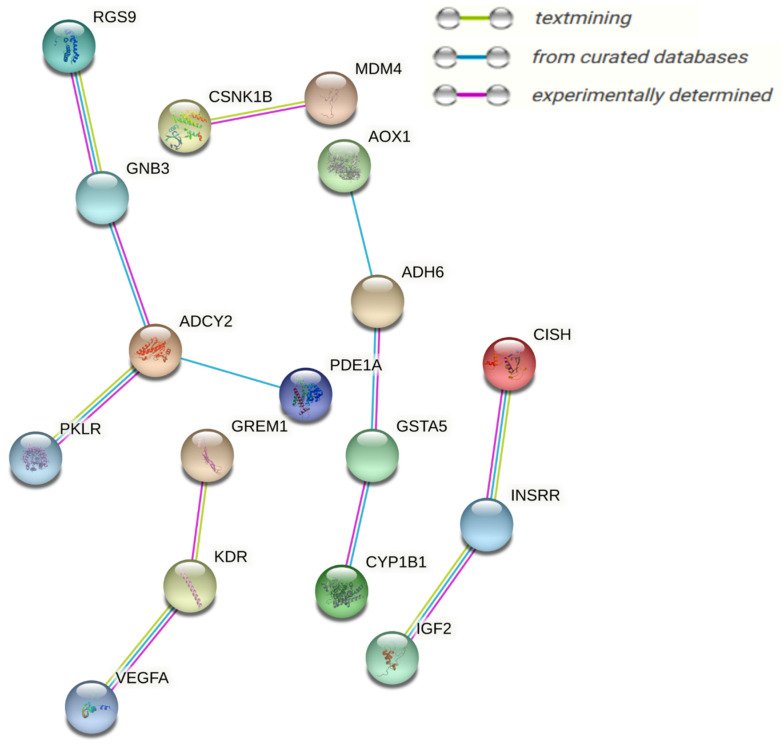
Differentially expressed genes found in multiple integrations and the network analysis of the transcriptome and metabolome response of granulosa cells to acute heat stress are drawn into a physical protein–protein interaction network (STRING); medium confidence score (0.4) nodes represent proteins and lines between nodes, referring to edges showing various sorts of interactions, denoted by different colors and defined through the figure legends.

**Table 1 biology-11-00839-t001:** Statistical summary of the top 25 metabolic and cellular pathways found in the integrated joint pathway analysis of significantly differentially expressed genes and important metabolites found in heat-stressed granulosa cells.

Pathways	Total /Hits	Raw *p*-Value	Holm Adjust *p*-Value
ABC transporters	198/18	5.1 × 10^−11^	1.7 × 10^−8^
Central carbon metabolism in cancer	103/11	6.5 × 10^−8^	2.1 × 10^−5^
Aminoacyl-tRNA biosynthesis	118/10	2.3 × 10^−6^	7.4 × 10^−4^
Protein digestion and absorption	168/11	8.7 × 10^−6^	2.8 × 10^−3^
Glycine, serine, and threonine metabolism	93/8	2.2 × 10^−5^	7.0 × 10^−3^
Mineral absorption	84/6	6.5 × 10^−4^	2.1 × 10^−1^
Vitamin B6 metabolism	36/4	1.0 × 10^−3^	3.3 × 10^−1^
Vitamin digestion and absorption	65/5	1.3 × 10^−3^	4.3 × 10^−1^
Relaxin signaling pathway	136/7	1.6 × 10^−3^	5.3 × 10^−1^
Dilated cardiomyopathy (DCM)	102/6	1.8 × 10^−3^	5.7 × 10^−1^
Arginine biosynthesis	42/4	1.9 × 10^−3^	5.9 × 10^−1^
Taste transduction	112/6	2.9 × 10^−3^	9.1 × 10^−1^
Arrhythmogenic right ventricular cardiomyopathy (ARVC)	78/5	3.0 × 10^−3^	9.4 × 10^−1^
Tyrosine metabolism	117/6	3.6 × 10^−3^	1.00000
Phenylalanine metabolism	83/5	3.9 × 10^−3^	1.00000
Cocaine addiction	56/4	5.3 × 10^−3^	1.00000
Tryptophan metabolism	129/6	5.8 × 10^−3^	1.00000
Glyoxylate and dicarboxylate metabolism	92/5	6.1 × 10^−3^	1.00000
TGF-beta signaling pathway	93/5	6.3 × 10^−3^	1.00000
Hypertrophic cardiomyopathy (HCM)	95/5	6.9 × 10^−3^	1.00000
Axon guidance	180/7	7.7 × 10^−3^	1.00000
GABAergic synapse	99/5	8.2 × 10^−3^	1.00000
Alanine, aspartate and glutamate metabolism	65/4	9.0 × 10^−3^	1.00000
Apelin signaling pathway	149/6	1.1 × 10^−2^	1.00000
Histidine metabolism	70/4	1.2 × 10^−2^	1.00000

**Table 2 biology-11-00839-t002:** Important functional pathways, metabolites, and genes. Important enriched pathways from an integrated analysis of metabolites and genes are divided into the top general functional pathways, top cell signaling and functional pathways, and cancer pathways. Complete statistical detailing of pathways and a list of functional hits along with their regulation statuses are given.

Joint Pathways	Total /Hits	Raw *p*-Value	Holm Adjust *p*-Value	Metabolites	Genes
**Top functional pathways from the integrated analysis**					
ABC transporters	198/18	5.11 × 10^11^	1.67 × 10^−8^	↓Alpha-Trehalose; ↓D-Mannitol; ↑L-Lysine; ↑L-Arginine; ↑L-Glutamine; ↓L-Histidine; ↑L-Leucine; ↓L-Threonine; ↓Proline; ↑Choline; ↓Thiamine; ↓Ciliatine; ↓L-Phenylalanine; ↓Riboflavin; ↓Uridine; ↓Xanthosine	↓ABCC5; ↓ABCA2;
Protein digestion and absorption	168/11	8.70 × 10^−6^	0.002819	↑L-Leucine; ↓L-Phenylalanine; ↓L-Tryptophan; ↓L-Threonine; ↑L-Glutamine; ↑L-Arginine; ↑L-Lysine; ↓L-Histidine; ↓Proline; ↑L-Tyrosine	↓COL11A2
Taste transduction	112/6	0.0028879	0.91259	↓Serotonin; ↑Citric Acid	↓SCN9A; ↓PDE1A; ↓GABBR1; ↓GNB3
Cocaine addiction	56/4	0.0053408	1.00000	↑L-Tyrosine;	↓MAOB; ↓RGS9; ↓GRIN2D
GABAergic synapse	99/5	0.0082127	1.00000	↑L-Glutamine; ↓Succinic Acid	↓GABBR1; ↓ADCY2; ↓GNB3
**Functional signaling pathways (manual query of integrated analysis)**					
cAMP signaling pathway	254/8	0.015042	1.00000	↓Serotonin; ↓Succinic Acid	↓RYR2; ↓ATP2A1; ↓GRIN2D; ↓GABBR1; ↓ADCY2; ↓OXT
Ovarian steroidogenesis	78/4	0.016783	1.00000	↓Progesterone	↓ADCY2; ↓CYP1B1 ↓PLA2G4B
mTOR signaling pathway	158/4	0.1386	1.00000	↑L-Leucine; ↑L-Arginine	↓DEPTOR; ↓WNT11
AMPK signaling pathway	146/2	0.5515	1.00000	↓AICAR	↑SLC2A4
**Cancer pathways (manual query of integrated analysis)**					
Central carbon metabolism in cancer	103/11	6.5 × 10^−8^	2.1 × 10^−5^	↑L-Glutamine; ↑Citric Acid; ↓Succinic Acid; ↑L-Leucine; ↓L-Phenylalanine; ↓L-Histidine; ↓L-Tryptophan; ↑L-Tyrosine; ↓Proline; ↑L-Arginine	↑SLC16A3
Breast cancer	153/3	0.30314	1.00000	↓Progesterone	↓CSNK1B; ↓WNT11
Prostate cancer	109/2	0.40000	1.00000	↓Progesterone	↓INSRR
Pathways in cancer	573/8	0.43410	1.00000	↓Progesterone	↓WNT11; ↓IGF2; ↑NOS2; ↓VEGFA; ↓ADCY2; ↓GNB3; ↑GSTA5

↑: up-regulated in heat stress; ↓: down-regulated in heat stress.

**Table 3 biology-11-00839-t003:** Important functional pathways, metabolites, and genes. Important metabolic pathways from an integrated analysis of metabolites and genes are presented. Complete statistical detailing of the pathways and a list of functional hits along with their regulation statuses are given.

Metabolic Pathways	Total /Hits	Raw *p*-Value	Holm Adjust *p*-Value	Metabolites	Genes
Vitamin B6 metabolism	21/4	0.00156	0.12918	↑Pyridoxine; ↓Pyridoxal; ↑4-Pyridoxic Acid	↑AOX1
Glycine, serine, and threonine metabolism	72/7	0.00180	0.14759	↑Choline; ↓Glycocyamine; ↓L-Threonine; ↑L-Allo-Threonine;	↓MAOB; ↓AOC2; ↓AMT
Phenylalanine metabolism	24/4	0.00261	0.21157	↓L-Phenylalanine; ↑L-Tyrosine;	↓AOC2; ↓MAOB
Arginine biosynthesis	27/4	0.00408	0.32625	↑L-Arginine; ↑L-Glutamine;	↑NOS2; ↓GPT
Tryptophan metabolism	84/6	0.017185	1.00000	↓L-Tryptophan; ↓Serotonin; ↑Indole-3-Acetaldehyde;	↓CYP1B1 ↓MAOB; ↑AOX1
Arginine and proline metabolism	78/5	0.04371	1.00000	↑L-Arginine; ↓Glycocyamine; ↓Proline	↑NOS2; ↓MAOB
Histidine metabolism	32/3	0.04444	1.00000	↓Urocanic Acid; ↓L-Histidine	↓MAOB
Glyoxylate and dicarboxylate metabolism	56/4	0.04982	1.00000	↓Cis-Aconitic Acid; ↑Citric Acid; ↑L-Glutamine	↓AMT
Alanine, aspartate and glutamate metabolism	61/4	0.06459	1.00000	↑L-Glutamine; ↑Citric Acid; ↓Succinic Acid	↓GPT
Tyrosine metabolism	88/5	0.06715	1.00000	↑L-Tyrosine	↓MAOB; ↑ADH6; ↓AOC2; ↑AOX1
Starch and sucrose metabolism	43/3	0.09127	1.00000	↓Alpha-Trehalose	↓AMY2B; ↑PYGM
Purine metabolism	169/7	0.12673	1.00000	↓Xanthine; ↑L-Glutamine; ↓AICAR; ↓Xanthosine	↓PDE1A; ↓ADCY2; ↓PKLR
One carbon pool by folate	31/2	0.18155	1.00000	↑Folic Acid	↓AMT

↑: up-regulated in heat stress; ↓: down-regulated in heat stress.

**Table 4 biology-11-00839-t004:** Statistical details of the top interaction nodes found in the joint network analysis of genes and metabolites, along with their regulation statuses in heat-stressed granulosa cells. “Degree” denotes the number of direct connections between the nodes; “Betweenness” shows the centrality of the given interaction.

Id	Label	Regulation Status	Degree	Betweenness
C00410	Progesterone	Down	13	544.42
C00780	Serotonin	Down	9	254.42
1545	*CYP1B1*	Down	4	281
C00158	Citric acid	Up	4	145
5020	*OXT*	Down	4	112
C00250	Pyridoxal	Down	3	110
3383	*ICAM1*	Up	2	170
C00299	Uridine	Down	2	170
5837	*PYGM*	Up	2	140
3291	*HSD11B2*	Down	2	74
108	*ADCY2*	Down	2	51
4129	*MAOB*	Down	2	38
7422	*VEGFA*	Down	2	38
5313	*PKLR*	Down	2	38
6517	*SLC2A4*	Up	2	38
2875	*GPT*	Down	2	38
C00695	Cholic acid	Down	2	38
4843	*NOS2*	Up	2	12.5
43	*ACHE*	Down	2	6.5
C00114	Choline	Up	2	6.08
C00047	L-Lysine	Up	2	3.08

**Table 5 biology-11-00839-t005:** Major diseases and their directly associated metabolites and genes are listed. “Degree” denotes the number of direct connections between nodes, and “Betweenness” shows the centrality of the given interaction. The list of interaction network nodes is given against each corresponding disease along with its regulation status in the heat-stressed granulosa cells.

Id	Label	Regulation Status	Degree	Betweenness
C00158	Citric acid	Up	131	26,439.49
C00047	L-Lysine	Up	112	17,872.77
C00042	Succinic acid	Down	105	20,430.26
C00064	L-Glutamine	Up	105	17,053.91
C00780	Serotonin	Down	101	20,433.61
C00123	L-Leucine	Up	76	7023.93
C00188	L-Threonine	Down	72	5723.68
C00082	L-Tyrosine	Up	65	9222.24
C00114	Choline	Up	64	9738.79
C00148	L-Proline	Down	56	5378.26
C05519	L-Allo-threonine	Up	52	3135.83
C00299	Uridine	Down	50	9847.77
C00695	Cholic acid	Down	41	4475.47
C00250	Pyridoxal	Down	39	2385.19
C00984	D-Galactose	Down	33	6225.15

**Table 6 biology-11-00839-t006:** Major diseases and their directly-associated metabolites and genes from the network analysis are listed. “Degree” denotes the number of direct connections among nodes and “Betweenness” shows the centrality of the given interaction. The list of interaction network nodes is given against each corresponding disease along with its regulation status in the heat-stressed granulosa cells.

Id	Disease	Degree	*b*/*w*	Metabolites	Genes
181500	SCHIZOPHRENIA	10	1086	↓Progesterone, ↑L-Lysine, ↓Serotonin, ↑Citric acid, ↑L-Glutamine, ↓Threonine, ↓Aminoadipic acid	↓*VEGFA*
104300	ALZHEIMER DISEASE	8	542	↑L-Lysine, ↑L-Leucine, ↑L-Glutamine, ↑Choline, ↓L-Proline, ↓Threonine	-
616299	LIPOYLTRANSFERASE 1 DEFICIENCY	5	286	↑L-Lysine, ↑L-Leucine, ↑L-Glutamine, ↓Succinic acid, ↓L-Proline, ↓Threonine	-
271900	CANAVAN DISEASE	4	453	↓Uridine, ↓Xanthine, ↑Citric acid, ↓Succinic acid	-
617290	EPILEPSY, VITAMIN B6-DEPENDENT	3	203	↓Pyridoxal, ↑Pyridoxine, ↑L-Leucine	-
276700	TYROSINEMIA, TYPE I	3	56	↓Threonine, ↑L-Tyrosine, ↑L-Lysine	-

*b*/*w*: betweenness; ↑: up-regulated in heat stress; ↓: down-regulated in heat stress.

## Data Availability

All pertinent data, including that of RNA-seq and metabolomics, are already reported or presented in the manuscript and associated Appendix A.

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
