# Peer review of "Joint Transcriptome and Metabolome Analysis Prevails the Biological Mechanisms Underlying the Pro-Survival Fight in In Vitro Heat-Stressed Granulosa Cells"

_biology, 2022, doi:10.3390/biology11060839_

Round 1

Reviewer 1 Report

Thank you for answering my queries. I accept this manuscript for publication.

Author Response

Thank you very much for a contructive peer-review of this article. Revised version of the manuscript is improved.

Reviewer 2 Report

The authors addressed most of the points raised, and this is fine now. Yet the un-physiological situation of exposure of ovarian cells to 43 C remains an issue. Lower temperatures would have been much more instructive. The temperature studied should be mentioned in the abstract.

Author Response

Thank you very much for a constructive peer-review of this article. Heat stress treatment temperature has been mentioned in the revised abstract part. The revised version of the manuscript is much improved.

Reviewer 3 Report

Figure 1 legend not described appropriately and systematically. Immunofluorescence images need field of view or micrograph measurement; optical zoom is not the standard unit for description.

Also why are the results of figure 1 (published previously) presented as the results of the current paper?? This can used for discussion but not used as a main figure.

English writing does not meet the standards for publications in Biology. At many times, statements written as such do not make any sense.

Poor discussion of the results.

Abbreviations not described appropriately.

The PCA analysis of the transcriptome of heat-stressed granulosa cells are highly variable (T2 is distinctly clustered away from T1 and T3). N=3 is also on the low. So, the experiment needs more comparable replicates (n=5).

Treatment groups need better explanation.

The method for apoptosis assay needs better and more detailed description for readers to understand. Explanation of how to determine early vs late apoptosis is required. Asking readers/reviewers to go read another paper to understand the current paper is not appropriate.

Line 350-351: Description for how cell proliferation, if performed as stated in this study, was determined is missing in methods section.

The DEGs separated by up- and down-regulated genes will give better readability and understanding to readers.

Ln 365: fold change or logFC?

Author Response

Thank you very much for a constructive peer-review of this manuscript. 

Please see the attached response letter to your comments.

Round 2

Reviewer 2 Report

The authors made an effort to address the points of concern.

Author Response

Thank you very much!

Reviewer 3 Report

The authors have done a good job addressing the previous comments. However, the following suggested edits before publication would improve the quality of the manuscript. 

Ø  Interchange the positioning of Figure 1 C and F for better readability.

Ø  The legend of X-axis for Figure 1 B, C, F, G should read Control and HS as this has been defined earlier in the manuscript and in Figure 1 A.

Ø  Line 77: replace treatment group with heat-stressed GCs compared to that of control GCs.

Ø  Ln 87: consider replacing represented with presented

Ø  Figure 1F: Why use a and c but not a and b to indicate statistical significance?

Ø  Line 69: consider adding in vitro heat stress to the title.

Ø  Line 94-96: Cite the referenced studies.

Ø  Line 97 -99: Use this- The list 97 of significantly differentially expressed genes and metabolites along with their logFC values used in this study is provided in supplementary table S1.

Ø  Line 242-243: In complete sentence. Should not start a sentence with where.

Ø  Line 246: Be consistent in using hrs or h

Ø  Line 248-250: Consider repharsing to Interestingly, our previous report indicates that the changing the cell culture medium immediately after heat stress was too early to show any effects on cell proliferation compared to that when changing the cell culture medium at 48 h after treatment.

Ø  Line 250-251: This phenomenon needs further investigation; a preliminary

Author Response

Thank you for the worthy reviewer's kind suggestions for further improvement of the manuscript presentation. All of the recent suggestions are accepted by the authors and corresponding changes are highlighted in red in the revised version of the manuscript.

This manuscript is a resubmission of an earlier submission. The following is a list of the peer review reports and author responses from that submission.

Round 1

Reviewer 1 Report

Dear Authors,

             This manuscript entitled"Joint transcriptome and metabolome analysis prevails biological mechanisms underlying the prosurvival fight in heat-stressed granulosa cells" is well designed research to understand the effect of heat stress on bovine granulosa cells employing transcriptomics and metabolomics techniques. Even though this study is novel and provides insight into the mechanism associated with heat stress in granulosa cells but there are a few concerns that needs to be addressed.

  1. It was mentioned that the samples (ovaries) were collected from local abattoir, so is it possible to mention the age of the animals (1-3years, 2-5 etc) as samples (granulosa cells) from the older animal are more prone to apoptosis and cell death compared to younger animal.
  2. What percentage of thecal cell contamination was found in cells cultured (ratio is also fine)?Because pathways related to ovarian steroidogenesis is affected due to the presence of thecal cell contamination. 
  3. Is it possible to include estrogen hormone levels from culture media as supplementary data to support the purity of granulosa cells in culture?
  4. Introduction can be improved by providing some more recent studies.
  5. Discussion should be improved so that mechanisms related to oxidative stress, steroidogenesis, apoptosis due to acute heat stress should provide more new insight into already existing mechanisms.

Best Regards    

Reviewer 2 Report

Dear authors,

Thanks for submitting your manuscript to our journal. We have the following comments on your manuscript:

Your manuscript is not well-explained. For example, in the result part, the manuscript did not explain what the discoveries were and what the discoveries meant. The data used in the manuscript was generated for your previous publications, although you did combine those data to do bioinformatics analysis. In addition, apart from showing what the KEGG analysis was, you did not explain what the analysis results meant. Last but not least, the manuscript did not have any validation data to support the analysis. Therefore, after consideration, we can't consider your manuscript to be published in our journal at this moment.

Detailed comments:

Using the data showing the transcriptional profile and metabolome profile you collected, this manuscript combined these two sets of data and did the bioinformatics analysis. By doing so, the manuscript aimed to identify important genes and metabolites which are important for granulosa cells to respond to heat stress. Moreover, the signaling pathways, the metabolic pathways, and the interactions among these genes/metabolites were also identified from the bioinformatics analysis.   The manuscript did provide information to help us understand how the granulosa cells respond to heat stress. Compared to other literatures which characterize the mechanism of granulosa cells responding to heat stress, the manuscript did provide more comprehensive knowledge for the research field by combining the transcriptional profile and metatolome profile. However, the manuscript just listed the analysis results without indicating what these results mean and how these results can help us for the disease treatment etc. In addition, the result part of the manuscript is not well-explained. The result part did not elaborate what data/results the bioinformatics analysis got, except for some brief descriptions in the figure legends. Last but not least, all of these bioinformatics analysis did not have experimental data to validate the discoveries. Therefore, considering all of these factors, I do not think level of this manuscript reaches to the level for publishing at MDPI-biology journal at this moment.

Thanks for understanding! Good luck with your manuscript!

Reviewer 3 Report

I am afraid to say that this paper does not provide the key information needed to be able to evalaute it.

There is no information on the source and nature of the granulosa cells! Animal? Human? Which follicular stage? How many? No information on the method (temperature, time..). An no information on the biological meaning of doing such studies in the first place.

My suggestion to reject has two main reasons:   1. I am afraid I simply can not evaluate the paper:     The authors wrote (4. Methods): …"This study did joint integration analysis of our previous metabolome [22] and transcriptome data from in-vitro 247 acute heat stressed (43 °C for 2 h) bovine granulosa cells. Both of those earlier omics studies involved same samples and 248 replicates (n = 3)…."   Thus crucial information is missing: For example,  what follicle stage or stages are the cells from, how was the quality (vitality, contamination with other cells..) of the cells assessed? What does n = 3 mean? How old were the cows, dairy cows, others? When/how was the " heat stress“ applied (fresh cells, cultured cells, what was the purity of the cells in the first place). Why 43 C and 2 h?   Without such information, the analyses are not of much value. Such information must be part of this paper and not part of other papers.   2. Another important point is the general biological relevance of the study. It is not - or not adequately - addressed.     For example, is there evidence that cows when exposed to 43 C (i.e. high fever  but for 2 h only?) show alterations of ovarian functions that are related to the outcome of this study? The authors start the discussion with citing  Kovats, R.S.; Hajat, S. Heat stress and public health: A critical review. In Proceedings of the Annual Review of Public Health; 2008; Vol. 29. This review deal with heat as an environmental and occupational hazard. This is not  - to my opinion -  related to 43 C body temperature (i.e. high fever). Citing it here is a long stretch and not adequate.    They continue with citing Gaskins, A.J.; Mínguez-Alarcón, L.; VoPham, T.; Hart, J.E.; Chavarro, J.E.; Schwartz, J.; Souter, I.; Laden, F. Impact of ambient temperature on ovarian reserve. Fertil. Steril. 2021, 116, doi:10.1016/j.fertnstert.2021.05.091. Again, high ambient temperatures are not related to 43 C body, or ovarian temperature.    Another study, which they cite ( Bridges PJ, Brusie MA, Fortune JE. Elevated temperature (heat stress) in vitro reduces androstenedione and estradiol and increases progesterone secretion by follicular cells from bovine dominant follicles. Domest Anim Endocrinol. 2005 Oct;29(3):508-22. doi: 10.1016/j.domaniend.2005.02.017. Epub 2005 Mar 14. PMID: 16153500) studied maximally 41 C, not 43 C.   The authors write as (5) Conclusion:  …."The etiology of various changes in heat stressed granulosa can be at least in part attributed to varying degree of cellular stress, impairment in lipid transport and metabolism, inflammation-metabolism nexus, and disruption in energetic metabolism. Besides, the demonstration of various cell signaling and metabolic pathways, this integration analysis of metabolome and transcriptome acute heat stressed granulosa cell provided an interesting conjecture of metabolites and genes through integrated analysis of transcriptome and metabolome data….“    This is not a conclusion but summarizes the cellular experiments (which I can not judge in the first place), and it does not address the biological relevance at all.    Thus, I am afraid I can not recommend anything other that to reject.  I would like to suggest, though that the authors may be given a chance to address the points and, possibly, resubmit.